# Distribution, Antibiotic Resistance, and Virulence Factors of *Vibrio parahaemolyticus* in the Southern Coastal Waters of Republic of Korea

**DOI:** 10.3390/antibiotics14050435

**Published:** 2025-04-26

**Authors:** Hyunwoo Zin, Intae Ham, Soonbum Shin, Hongsik Yu, Tae-Jin Choi, Kwangsoo Ha, Jong Soo Mok

**Affiliations:** 1Food Safety and Processing Research Division, National Institute of Fisheries Science, Busan 46083, Republic of Korea; 2Department of Microbiology, Pukyong National University, Busan 48513, Republic of Korea

**Keywords:** *Vibrio parahaemolyticus*, antibiotic resistance, virulence genes

## Abstract

**Background/Objectives:** *Vibrio parahaemolyticus* is a marine bacterium and a major cause of food poisoning worldwide, primarily associated with gastric illnesses such as gastroenteritis. This study aimed to investigate the distribution, antibiotic resistance, and virulence genes of *V. parahaemolyticus* present in shellfish and seawater of the southern coast of Korea, a major shellfish harvesting area. **Methods:** Shellfish and seawater samples were collected monthly in 2023 from 24 coastal sites in Korea. *V. parahaemolyticus* was isolated and identified using the MPN method, biochemical tests, MALDI-TOF mass spectrometry, and 16S rRNA sequencing. Antimicrobial susceptibility was tested for 673 isolates using the Sensititre MIC system, and virulence genes (*tdh* and *trh*) were detected by PCR. **Results:**
*V. parahaemolyticus* had a detection rate of 18.2–58.3% in shellfish and 8.3–50% in seawater samples. Among the isolates, 97.9% and 97.3% were resistant to ampicillin and colistin, respectively, while 8.3% showed resistance to four or more antibiotics. The virulence genes *tdh* and *trh* were detected in 0.45% and 3.34% of shellfish samples and 1.23% and 4.46% of seawater samples, respectively. **Conclusions:** These findings will help implement appropriate precautionary measures to prevent potential human health risks arising from exposure to multidrug-resistant or pathogenic *V. parahaemolyticus*.

## 1. Introduction

*Vibrio* is a genus of globally distributed Gram-negative bacteria primarily found in marine environments [1]. Changes in seawater temperature and salinity are key factors in the expansion of the *Vibrio* range [2]. These environmental shifts significantly affect bacterial survival, pathogenicity, and gene expression. For instance, rising water temperatures promote the growth and reproduction of *Vibrio*, whereas variations in salinity may enhance its adaptability and resistance, further contributing to its spread [3,4,5]. Recent climate changes, such as global warming and fluctuations in sea temperatures, have accelerated the spread of *Vibrio* strains [6].

Pathogenic *Vibrio* strains can cause gastrointestinal illnesses in humans—e.g., acute gastroenteritis when consuming contaminated seafood, such as undercooked or raw shellfish and fish [7]. The pathogenicity of *Vibrio parahaemolyticus* is primarily linked to hemolysin proteins encoded by the thermostable direct hemolysin (*tdh*) and TDH-related hemolysin (*trh*) genes [8]. These toxins disrupt host cell membranes by forming pores, leading to cell lysis, gastrointestinal inflammation, and symptoms like diarrhea and abdominal cramps [9]. While not all *V. parahaemolyticus* strains harbor these genes, their presence is strongly associated with heightened virulence and more severe infections. This issue is particularly prominent in countries with high seafood consumption during summer, making it a growing public health concern [10]. *V. parahaemolyticus* is one of the leading causes of *Vibrio* infections globally and is frequently reported in regions such as Asia and North America, particularly during summer when water temperatures rise [4,7,11]. South Korea has observed an increase in cases of gastroenteritis caused by *Vibrio* linked to the consumption of raw or undercooked seafood [12]. In response, *Vibrio* monitoring and surveillance efforts have been enhanced [4].

In addition to the rising infection rates, another concerning issue associated with the *Vibrio* species is their increasing antibiotic resistance in marine environments [13,14]. Antibiotics are often overused for infection prevention and treatment purposes [15]. This overuse was amplified during the COVID-19 pandemic, which led to widespread antibiotic use and accelerated the spread of resistant bacteria [16,17]. Furthermore, the contamination of marine and agricultural ecosystems with antibiotics has exacerbated the transfer of antibiotic resistance genes to *Vibrio* species [18]. Therefore, antibiotic resistance has emerged as a significant public health concern in various countries [19].

Pathogenic bacteria, along with their resistance genes, are often released into aquatic systems through wastewater [20]. This process allows naturally occurring marine bacteria, particularly *Vibrio*, to serve as reservoirs for resistance genes, playing a crucial role in the evolution and spread of antibiotic resistance in these environments [21,22]. This phenomenon represents a severe risk to both human health and the seafood industry, particularly in the case of multidrug-resistant *Vibrio* strains [23,24].

Understanding the occurrence and antibiotic susceptibility patterns of major pathogenic *Vibrio* species in marine environments is crucial for seafood safety and public health management. While previous studies in Korea have investigated *Vibrio parahaemolyticus*, many were conducted within a limited number of sampling sites, constraining the spatial generalizability of their findings. To address this limitation, the present study conducted a year-round survey across 24 shellfish harvesting areas along the southern coast—a region responsible for the majority of shellfish production in Korea—analyzing both seawater and shellfish samples to evaluate ecological distribution patterns, antimicrobial resistance, and seasonal variation in key virulence genes (*tdh* and *trh*). This broader spatial coverage provides a comprehensive foundation for evaluating seafood safety risks and contributes valuable insights for strengthening future public health monitoring and food safety measures in Korea. Although the study focuses on Korean coastal waters, the findings may also serve as a useful reference for other regions experiencing similar environmental conditions, offering comparative value for global research on marine pathogens and antimicrobial resistance.

## 2. Results

### 2.1. Water Temperature and Salinity

The monthly variations in seawater temperature and salinity measured along the Korean coast in 2023 are presented in Figure 1. Seawater temperatures ranged from 7.6 °C in February (SD ± 1.8 °C) to 26.3 °C in August (SD ± 2.5 °C), reflecting significant seasonal variability typical of Korea’s four-season climate. By contrast, salinity remained relatively stable, with monthly averages ranging from 27.05 PSU in August to 33.49 PSU in March. A temporary drop in salinity during the summer was likely caused by increased rainfall during the rainy season.

### 2.2. Distribution of V. parahaemolyticus

The prevalence of *V. parahaemolyticus* in 411 analyzed samples is summarized in Table 1. In shellfish, concentrations ranged from <30 MPN/100 g to 11,000 MPN/100 g, with the highest monthly average of 6011 MPN/100 g observed in August. Detection rates increased from 6.7% in April to 100% in August (Figure 2a). In seawater, concentrations ranged from <3.0 MPN/100 mL to 1100 MPN/100 mL, with a peak average of 248 MPN/100 mL in August. Detection rates in seawater rose from 5.3% in May to 84.2% in August (Table 1, Figure 2a). Shellfish showed a higher overall detection rate (41.1%) compared to seawater (27.4%).

### 2.3. Antimicrobial Resistance Profiles of V. parahaemolyticus

Antimicrobial resistance testing of 673 *V. parahaemolyticus* isolates from shellfish and seawater was conducted using 16 antibiotics (Figure 3). Resistance to ampicillin (AMP) and colistin (CL) was most prevalent, observed in 97.9% and 97.3% of isolates, respectively. Resistance to sulfisoxazole (FIS) and streptomycin (STR) was also notable, at 26.2% and 22.3%. Resistance rates to other antibiotics, such as FOX, SXT, and TET, were significantly lower, ranging from 0.89% to 1.04%. Importantly, all isolates were fully susceptible to ceftazidime (CAZ), ciprofloxacin (CIP), gentamicin (GEN), and nalidixic acid (NAL). The MAR index, an indicator of multidrug resistance, ranged from 0.00 to 0.38, with 8.3% of isolates exceeding the threshold of 0.2 (Table 2). These findings highlight the potential for multidrug resistance in marine environments and align with global reports of antibiotic resistance in *V. parahaemolyticus* strains [22,25,26].

### 2.4. Abundance Variations and Virulence Genes of V. parahaemolyticus in Shellfish and Seawater

The detection rates of *V. parahaemolyticus* in shellfish samples ranged from 18.2% to 58.3%, with higher rates at coastal sites such as points 1, 4, and 12 (Figure 4A). In seawater, detection rates ranged from 8.3% to 50%, with elevated levels observed at points 16, 17, and 18 (Figure 4B). Table 3 summarizes the detection of virulence genes, with *tdh* identified in 0.45% of shellfish samples and 1.23% of seawater samples and *trh* detected in 3.34% and 4.46% of shellfish and seawater samples, respectively.

## 3. Discussion

This study provides foundational data on the ecological distribution, antimicrobial resistance, and virulence gene profiles of *Vibrio parahaemolyticus* in Korean coastal waters. Although the study does not aim to fully assess ecological risks, the findings offer critical baseline information for future research exploring the interactions between environmental factors and pathogenic characteristics. Seasonal variations in seawater temperature and salinity observed in this study underscore the significant influence of environmental factors on marine ecosystem dynamics and the distribution of *V. parahaemolyticus.*

Seawater temperatures ranged from 7.6 °C in February to 26.3 °C in August, consistent with Korea’s distinct four-season climate. These findings align with previous studies that highlighted temperature as a key driver of marine ecosystem dynamics and pathogen distribution [20,27,28,29,30]. In contrast, salinity remained relatively stable, with a temporary drop during the summer, likely due to increased rainfall. While temperature is the primary factor influencing the distribution of *Vibrio* species, localized salinity changes may create microenvironments that facilitate bacterial growth, further emphasizing the need for continuous environmental monitoring, particularly during warmer months when the proliferation of *V. parahaemolyticus* is most pronounced.

The seasonal prevalence of *V. parahaemolyticus* observed in this study demonstrates the significant impact of seawater temperature on the dynamics of this pathogen. The high detection rates during warmer months, particularly in August, highlight the role of shellfish as reservoirs for *V. parahaemolyticus*, likely due to their filter-feeding mechanisms that concentrate bacteria. These results are consistent with global studies that report similar seasonal patterns and regional differences in *V. parahaemolyticus* prevalence. For example, research from China documented diverse serotypes and resistance profiles in clinical and environmental isolates [31]. Similarly, studies from the Gulf of Mexico identified distinct genetic lineages, indicating significant geographical differentiation [32]. The seasonal trends observed in Korea align with global patterns, underscoring the importance of regional monitoring to understand global pathogen distribution and evolution [27].

The antimicrobial resistance profiles observed in this study raise significant public health concerns. Resistance to ampicillin (97.9%) and colistin (97.3%) was notably high, consistent with global trends [19,33,34]. Ampicillin resistance in *V. parahaemolyticus* has been reported as an intrinsic trait, mediated by the CARB-17 family of β-lactamases encoded chromosomally [35]. In contrast, the mechanism underlying colistin resistance in this species is less clearly defined; however, recent studies suggest that modifications of lipid A in the bacterial outer membrane, such as those mediated by the *pmr* operon or *mcr*-like genes, may contribute to reduced susceptibility [36,37]. Notably, this study did not aim to experimentally distinguish whether the observed colistin resistance was intrinsic or acquired. Resistance to sulfisoxazole (26.2%) and streptomycin (23.3%) further suggests that the frequent use of these antibiotics in aquaculture may contribute to the emergence of resistant strains in marine environments. While ampicillin resistance is considered an intrinsic characteristic of *V. parahaemolyticus* [35], the notably high resistance to colistin observed in this study is of greater concern, as colistin is regarded as a last-resort antibiotic in clinical settings. This finding raises questions about potential environmental drivers and misuse of critically important antimicrobials, warranting stricter oversight of antibiotic application in aquaculture systems [38]. Furthermore, these findings highlight the importance of implementing routine surveillance of antimicrobial resistance in aquaculture environments. This study focused on aquaculture sites to reflect actual seafood production and consumer exposure; however, the absence of non-aquaculture control sites limits broader ecological comparisons.

The multidrug resistance (MDR) index in this study ranged from 0.00 to 0.38, with 8.3% of isolates exceeding the critical threshold of 0.2, often associated with clinical relevance. These findings underscore the necessity of stricter regulation of antibiotic use in aquaculture to limit the spread of antimicrobial-resistant strains. Horizontal gene transfer (HGT), including conjugation, transformation, and transduction, is considered a major mechanism contributing to the dissemination of antibiotic resistance genes among Vibrio species in aquatic environments [39,40]. The high resistance rates observed in this study may be partly attributed to horizontal gene transfer occurring within marine microbial communities.

Despite the presence of multidrug-resistant strains, all isolates in this study were fully susceptible to clinically important antibiotics, such as ceftazidime, ciprofloxacin, and gentamicin, which are recommended for treating *Vibrio* infections [41,42]. However, the widespread resistance to first-generation antibiotics like ampicillin highlights the importance of managing antibiotic use to prevent the spread of resistance and ensure the safety of seafood consumption. In Korea, where *V. parahaemolyticus* is a leading cause of seafood-related diseases, these findings emphasize the need for targeted environmental and shellfish monitoring during high-risk periods, particularly warmer months, along with public health advisories to mitigate seafood-borne illness risks.

These findings also support the need to strengthen seafood safety regulations by incorporating antibiotic resistance surveillance into routine monitoring frameworks and limiting the use of critical antimicrobials in aquaculture [38]. Regulatory policies should be adapted to include risk-based assessments and seasonal control measures, particularly during periods of high bacterial proliferation.

The detection of *tdh* and *trh* virulence genes further highlights the pathogenic potential of *V. parahaemolyticus* in Korea’s marine environment. These genes were selected due to their high epidemiological relevance and their role as key molecular markers for identifying pathogenic *V. parahaemolyticus* [8,43]. Although detection rates for *tdh* (0.45–1.23%) and *trh* (3.34–4.46%) were lower than those reported in other regions such as Chile, Taiwan, and the United States [4,44,45,46], the seasonal prevalence of these genes underscores the heightened risk of infections during warmer months. These differences may be attributed to variations in environmental conditions or sampling methodologies. The higher overall detection rate of *V. parahaemolyticus* in shellfish compared to seawater is consistent with the filter-feeding nature of bivalves, which enables them to accumulate both pathogenic and non-pathogenic strains. However, the higher detection rate of virulence genes in seawater observed in this study may reflect temporal fluctuations in the abundance of pathogenic strains in the water column. Since shellfish accumulate bacteria over time, their microbial profile may represent an averaged composition, potentially diluting the relative proportion of virulent strains. These differences highlight the complex dynamics of Vibrio populations between host organisms and their surrounding environment. Nevertheless, the presence of these virulence genes emphasizes the need for integrated monitoring of both shellfish and seawater, along with stringent food safety regulations, to prevent outbreaks of *V. parahaemolyticus*-associated illnesses.

In addition to surveillance efforts, it is also important to consider practical approaches to reduce the pathogenic risks associated with virulence gene expression. While *tdh* and *trh* are often described as “thermostable” hemolysins, they are proteinaceous in nature and can be effectively inactivated at temperatures above 70 °C [47]. Accordingly, thorough thermal processing remains a key intervention to reduce infection risks, as basic rinsing or shellfish cleaning does not sufficiently eliminate either the pathogen or its toxins [48]. Given these results, future studies should expand beyond *tdh* and *trh* to investigate additional virulence-related genes, such as those involved in the type III secretion system, and incorporate statistical modeling to better elucidate both the molecular mechanisms and environmental factors underlying the pathogenicity of *V. parahaemolyticus* and to inform more robust control strategies.

In conclusion, this study highlights the complex interplay between environmental factors, the prevalence of *V. parahaemolyticus*, and its resistance profiles in Korean coastal waters. The findings underscore the importance of continuous environmental monitoring, targeted public health interventions, and stricter regulation of antibiotic use to protect seafood safety and mitigate public health risks in regions with high seafood consumption.

## 4. Materials and Methods

### 4.1. Sample Collection

The samples for this study were collected monthly from January to December 2023 at 24 locations along the coasts of Gyeongsangnam-do, Jeollanam-do, and Busan, South Korea (Figure 5). Sampling points were selected along coastal regions with a high concentration of shellfish farms to represent areas with direct implications for public health. These locations were chosen based on their relevance to seafood production and potential risks associated with *Vibrio* contamination, reflecting real-world scenarios of human exposure. The samples included seawater, collected at all sampling sites, and shellfish collected only at designated sites. Shellfish species included oysters (*Crassostrea gigas*; points 1, 6, 7, 8, 10, 11, 12, 13, 19), mussels (*Mytilus galloprovincialis*; points 4, 5, 18), ark shells (*Scapharca subcrenata*; points 2, 20, 21), and Manila clams (*Ruditapes philippinarum*; point 22). Both seawater and shellfish samples were consistently collected during the same time period each month. In total, 411 samples were collected, comprising 226 seawater samples, 106 oysters, 33 mussels, 34 ark shells, and 12 Manila clams. All samples were processed following standardized sampling procedures and under controlled laboratory conditions, in accordance with the US Food and Drug Administration bacterial analysis manual [49]. The samples were transported under refrigeration (7–10 °C) and delivered to the laboratory within 8 h of collection. All microbiological analyses were initiated within 24 h to minimize the risk of bacterial proliferation or sample degradation. Seawater temperature and salinity were measured using a digital sampling system (ProDSS; YSI Incorporated, Yellow Springs, OH, USA).

### 4.2. Analysis of V. parahaemolyticus

Bacterial analysis was conducted immediately upon the arrival of samples at the laboratory. Shellfish samples (approximately 2 kg per site, including shells) were collected, and at least 10 individual shellfish were used to obtain 200 g of flesh meat and shell liquor for analysis. The samples were rinsed with tap water and shucked before analysis [49]. Both shellfish and seawater samples were quantified using the most probable number (MPN) method, with values calculated based on the presence of positive growth in a three-tube, three-replicate, ten-fold serial dilution. MPN values were determined following the standard methodology outlined in the Bacteriological Analytical Manual (BAM) [50], Appendix 2 [51], and the results were expressed as MPN/100 g for shellfish and MPN/100 mL for seawater. Shellfish flesh meat with liquor (200 g) was homogenized with 200 mL of phosphate-buffered solution (PBS; 2.5 mM KH_2_PO_4_, pH 7.2), and 20 g of the homogenate was further diluted 1:10 by adding 80 mL of PBS. For enrichment, 10 mL of 2× APW (2% NaCl, pH 8.5 ± 0.2) was inoculated with 10 mL of seawater or homogenized shellfish sample. Additionally, 10 mL of 1× APW was inoculated with 1 mL of seawater or 1 mL of a 10^−1^ dilution of homogenized shellfish, prepared by blending 200 g of flesh with an equal volume of PBS and further diluting 20 mL of the homogenate with 80 mL of PBS. All inoculated samples were incubated at 37 °C for 18–24 h. Two or three mauve colonies were typically selected from each positive plate, and up to 27 colonies per sample were isolated for downstream analysis, based on the three-tube, three-dilution MPN method (nine tubes per sample). After enrichment, 10 μL of each positive APW culture (both shellfish and seawater samples) was streaked onto CHROMagar™ Vibrio plates (CHROMagar, La Plaine St-Denis, France) and incubated at 37 °C for 24 h. Presumptive *V. parahaemolyticus* colonies (purple colonies) were subjected to further biochemical screening using oxidase reaction and triple-sugar iron agar fermentation tests. Subsequently, isolates were analyzed with MALDI-TOF mass spectrometry (Bruker Daltonics, Bremen, Germany; flexControl Version 3.4) for species-level identification. Calibration and quality control were performed using a bacterial test standard (BTS) solution. For positive control, Escherichia coli (KCTC 1682) was used, while the negative control consisted of a blank spot where 1 µL of matrix HCCA was dispensed. To minimize the risk of false negatives associated with MALDI-TOF, 16s rRNA gene sequencing was performed as a secondary confirmation using the 27F (5′-AGAGTTTGATCMTGGCTCAG-3′) and 1492R (5′-TACGGYTACCTTGTTACGACTT-3′) primers.

### 4.3. Antimicrobial Susceptibility Tests of V. parahaemolyticus Isolates

This study conducted antimicrobial susceptibility testing on 673 *V. parahaemolyticus* isolates by measuring the minimum inhibitory concentration (MIC). All isolates were initially inoculated onto tryptone soya agar (TSA, Chalco, Nebraska; OXOID, pH 7.3 ± 0.2) and incubated at 37 °C for 24 h. Thereafter, they were stored at −70 °C using cryobeads (Microbank; Pro-Lab Diagnostics Inc., Richmond Hill, ON, Canada) and reactivated as needed for testing. The MIC measurements were performed using the Sensititre MIC panel (KRNV5F panel; ThermoFischer Scientific, Waltham, MA, USA) according to manufacturer instructions. Bacterial colonies (3–5) grown on Mueller–Hinton agar (MilliporeSigma, Sigma-Aldrich, Burlington, MA, USA) at 37 °C for 18–24 h were picked and suspended in saline solution using a loop (0.85% NaCl). The turbidity of the suspension was adjusted to match the 0.5 McFarland standard. An aliquot of 10 μL of the adjusted suspension was dispensed into an 11 mL tube of Mueller–Hinton broth (Cation Adjusted Mueller–Hinton with TES zwitterionic buffering agent), and a dosing head was attached. The Sensititre MIC panel was secured in the plate holder of an autoinoculator (Sensititre AIM; ThermoFisher Scientific), and 50 μL of the suspension was dispensed into each well. The panel was sealed with film and incubated at 37 °C for 18–24 h. After incubation, the MIC values for each isolate were determined using an automated plate reading system (Sensititre OptiRead, ThermoFisher Scientific). The antimicrobial agents and concentration ranges included in the KRNV5F panel were as follows: amoxicillin/clavulanic acid (AmC, 2:1 ratio; 2/1–32/16 µg/mL), ampicillin (AMP; 2–64 µg/mL), cefepime (FEP; 0.25–16 µg/mL), cefoxitin (FOX; 1–16 µg/mL), ceftazidime (CAZ; 1–16 µg/mL), ceftiofur (XNL; 0.5–8 µg/mL), chloramphenicol (CHL; 2–64 µg/mL), ciprofloxacin (CIP; 0.12–16 µg/mL), colistin (CL; 2–16 µg/mL), gentamicin (GEN; 1–64 µg/mL), meropenem (MEM; 0.25–4 µg/mL), nalidixic acid (NAL; 2–128 µg/mL), streptomycin (STR; 16–128 µg/mL), sulfisoxazole (FIS; 16–256 µg/mL), tetracycline (TET; 2–128 µg/mL), and trimethoprim/sulfamethoxazole (SXT; 0.12/2.38–4/76 µg/mL). Resistance breakpoints for the above antimicrobials were interpreted primarily based on the Clinical and Laboratory Standards Institute (CLSI) M100 guidelines [52]. For streptomycin and ceftiofur, which are not included in the CLSI M100 standard, interpretive criteria were adopted from the National Antimicrobial Resistance Monitoring System [53]. These reference breakpoints are widely used in antimicrobial resistance surveillance and reflect standardized and internationally recognized criteria for assessing resistance phenotypes. Based on the MIC results, each isolate was categorized as resistant, intermediate, or sensitive to the tested antimicrobials. The multiple antibiotic resistance (MAR) index was calculated as x/y, where x represents the number of antimicrobials to which an isolate exhibited resistance and y is the total number of antimicrobials tested.

### 4.4. Virulence Genes in V. parahaemolyticus Isolates

The presence of virulence genes in the 673 isolates of *V. parahaemolyticus* was analyzed using polymerase chain reaction (PCR). The PCR was performed using a thermal cycler (VeritiPro Thermal Cycler; ThermoFisher Scientific), with the primer sets VPD-1 (5′-CCTTCCTGCCAACATTACAT-3′) and VPD-2 (5′-GGCTTCGATATTTTCAGTATCT-3′) for amplifying the *tdh* gene and VPR-1 (5′-TTGGCTTCGATATTTTCAGTATCT-3′) and VPR-2 (5′-CATAACAAACAT-ATGCCCATTTCCG-3′) for amplifying the *trh* gene [54]. PCR amplification followed the manufacturer’s protocol (Takara Bio Inc., Shiga, Japan) and was conducted under the following conditions: initial denaturation at 94 °C for 5 min, followed by 35 cycles of denaturation at 94 °C for 60 s, annealing at 55 °C for 60 s, and extension at 72 °C for 60 s, with a final extension at 72 °C for 5 min [54]. All amplified products were confirmed by electrophoresis and visualized using the QIAxcel Advanced system (Qiagen, Hilden, Germany). The expected product sizes were 251 bp for the *tdh* gene and 250 bp for the *trh* gene. Positive control templates VP1 and VP2 (Takara Bio Inc.) were used to validate the PCR performance, with sterile water serving as the negative control.

## 5. Conclusions

This study provides critical insights into the seasonal prevalence, antimicrobial resistance, and virulence profiles of *Vibrio parahaemolyticus* along the Korean coastline. The strong correlation between seawater temperature and the prevalence of *V. parahaemolyticus* highlights the seasonal risk posed by this pathogen, particularly during warmer months. The detection of virulence genes (*tdh* and *trh*) in both shellfish and seawater, coupled with the high prevalence of antibiotic-resistant strains, underscores a growing public health concern, especially regarding the consumption of raw or undercooked seafood.

These findings emphasize the urgent need for continuous surveillance of *V. parahaemolyticus* prevalence, environmental factors, and antimicrobial resistance to mitigate the risks associated with seafood consumption. Additionally, stricter regulation of antibiotic use in aquaculture and agriculture is essential to curb the emergence of multidrug-resistant strains.

Future studies should aim to explore the interaction between environmental factors, microbial resistance, and virulence to develop integrated management strategies. Such efforts will be instrumental in ensuring food safety and protecting public health in regions with high seafood consumption.

## Figures and Tables

**Figure 1 antibiotics-14-00435-f001:**
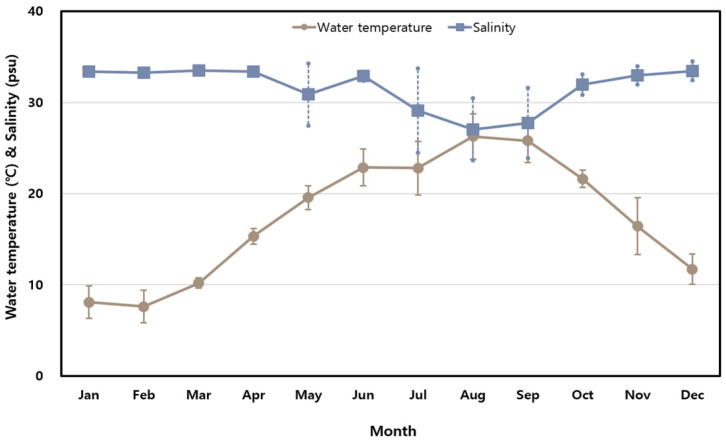
Monthly variations in seawater temperature and salinity at sampling stations.

**Figure 2 antibiotics-14-00435-f002:**
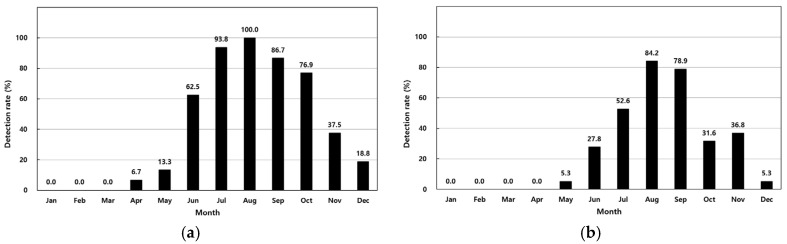
Monthly detection rates of *Vibrio parahaemolyticus* in shellfish (**a**) and seawater (**b**).

**Figure 3 antibiotics-14-00435-f003:**
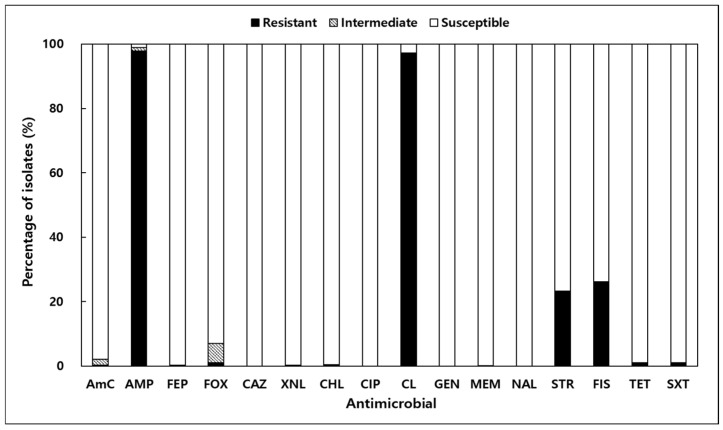
Antibiotic resistance results of *Vibrio parahaemolyticus* isolated from shellfish and seawater along the Korean coast; AmC, amoxicillin/clavulanic acid; AMP, ampicillin; CAZ, ceftazidime; CHL, chloramphenicol; CIP, ciprofloxacin; CL, colistin; FEP, cefepime; FIS, sulfisoxazole; FOX, cefoxitin; GEN, gentamicin; MEM, meropenem; NAL, nalidixic acid; STR, streptomycin; SXT, trimethoprim/sulfamethoxazole; TET, tetracycline; XNL, ceftiofur.

**Figure 4 antibiotics-14-00435-f004:**
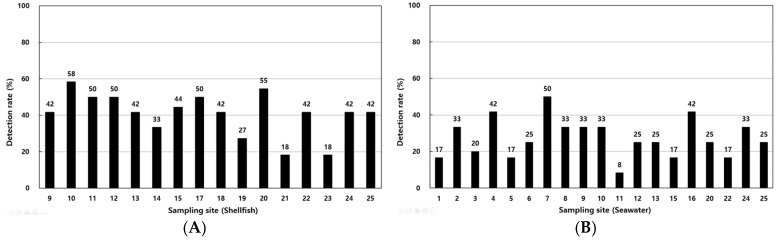
Annual detection rates of *Vibrio parahaemolyticus* in shellfish (**A**) and seawater (**B**) samples from various sampling sites along the Korean coast.

**Figure 5 antibiotics-14-00435-f005:**
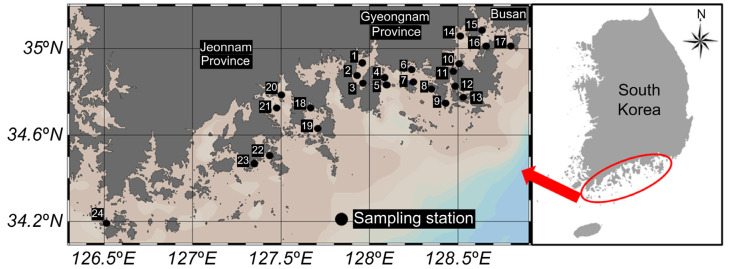
Sampling stations for shellfish included oysters (*Crassostrea gigas*) at points 1, 6, 7, 8, 10, 11, 12, 13, and 19; mussels (*Mytilus galloprovincialis*) at points 4, 5, and 18; ark shells (*Scapharca subcrenata*) at points 2, 20, and 21; and Manila clams (*Ruditapes philippinarum*) at point 22. Seawater samples were taken from points 1, 3, 6, 7, 8, 9, 10, 11, 12, 13, 14, 15, 16, 17, 18, 19, 22, 23, and 24.

**Table 1 antibiotics-14-00435-t001:** Monthly variations in *Vibrio parahaemolyticus* numbers in shellfish and seawater samples collected along the Korean coast.

Samples			Concentration (MPN/100 g or 100 mL)	Mean of Detected Values (MPN/100 g or 100 mL)
Month	Total Number	Positive Number		
Shellfish				
1	16	0	<30	<30
2	16	0	<30	<30
3	15	0	<30	<30
4	15	1	<30–36	36
5	15	2	<30–93	93
6	16	10	<30–1500	252
7	16	15	<30–11,000	3680
8	16	16	150–11,000	6011
9	15	13	<30–11,000	1433
10	13	10	<30–11,000	1695
11	16	6	<30–930	229
12	16	3	<30–62	42.7
Subtotal	185	76	<30–11,000	<30–6011
Seawater				
1	19	0	<3.0	<3.0
2	18	0	<3.0	<3.0
3	19	0	<3.0	<3.0
4	19	0	<3.0	<3.0
5	19	1	<3.0–3.0	3.0
6	18	5	<3.0–9.2	7.0
7	19	11	<3.0–460	100
8	19	16	<3.0–1100	248
9	19	15	<3.0–93	21.8
10	19	6	<3.0–9.2	4.3
11	19	7	<3.0–7.4	4.0
12	19	1	<3.0–3.6	3.6
Subtotal	226	62	<3.0–1100	<3.0–109

**Table 2 antibiotics-14-00435-t002:** MAR index values for *Vibrio parahaemolyticus* isolates (*n* = 673) from shellfish and seawater samples collected along the Korean coast.

Resistance Pattern	Number of Antimicrobials	Number of Isolates (%)	MAR Index
-	0	1 (0.15%)	0.00
AMP *	1	11 (1.63%)	0.06
CL		9 (1.34%)	0.06
AMP, CL	2	361 (53.64%)	0.13
AMP, FIS		4 (0.59%)	0.13
AMP, STR		1 (0.15%)	0.13
CL, STR		3 (0.45%)	0.13
AMP, CL, FIS	3	117 (17.38%)	0.19
AMP, CL, STR		104 (15.45%)	0.19
AMP, CL, SXT		1 (0.15%)	0.19
AMP, FEP, CL		1 (0.15%)	0.19
AMP, FOX, CL		2 (0.30%)	0.19
AMP, STR, FIS		1 (0.15%)	0.19
CL, STR, FIS		1 (0.15%)	0.19
AmC, AMP, CL, STR	4	1 (0.15%)	0.25
AMP, CL, FIS, SXT		2 (0.30%)	0.25
AMP, CL, FIS, TET		1 (0.15%)	0.25
AMP, CL, STR, FIS		41 (6.09%)	0.25
AMP, FOX, CL, FIS		3 (0.45%)	0.25
AMP, XNL, CL, STR		1 (0.15%)	0.25
AmC, AMP, FOX, CL, TET	5	1 (0.15%)	0.31
AMP, CL, STR, FIS, SXT		1 (0.15%)	0.31
AMP, CL, STR, FIS, TET		1 (0.15%)	0.31
AMP, FOX, CL, STR, FIS		1 (0.15%)	0.31
AMP, CHL, CL, FIS, TET, SXT	6	2 (0.30%)	0.38
AMP, CL, STR, FIS, TET, SXT		1 (0.15%)	0.38

* AMP, ampicillin; AmC, amoxicillin/clavulanic acid; CHL, chloramphenicol; CL, colistin; FEP, cefepime; FIS, sulfisoxazole; FOX, cefoxitin; MAR, multiple antimicrobial resistance; STR, streptomycin; SXT, trimethoprim/sulfamethoxazole; TET, tetracycline; XNL, ceftiofur.

**Table 3 antibiotics-14-00435-t003:** Distribution and pathogenic genes of *Vibrio parahaemolyticus* isolates from molluscan shellfish and seawater and seawater samples along the Korean coast.

Samples			Isolates	
Type	Positive Number/Total Number (%)	Concentration (MPN/100 g 100 mL)	Total Number	Positive Number for Pathogenic Genes
Molluscan shellfish				
Oyster	47/106 (44.3%)	<30–11,000	254	*tdh* (2), *trh* (13)
Mussel	15/33 (45.5%)	<30–11,000	108	*tdh* (0), *trh* (0)
Ark shell	5/12 (41.7%)	<30–11,000	33	*tdh* (0), *trh* (2)
Manila clam	9/34 (26.5%)	<30–1500	34	*tdh* (0), *trh* (0)
Subtotal	76/185 (41.1%)	<30–11,000	429	*tdh* (2), *trh* (15)
Seawater	62/226 (27.4%)	<3.0–1100	244	*tdh* (3), *trh* (10)
Total	138/411 (33.6%)	-	673	30 (*tdh*, 5; *trh*, 25)

## Data Availability

The original contributions presented in this study are included in the article. Further inquiries can be directed to the corresponding author.

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
