# Peer review of "Distribution, Antibiotic Resistance, and Virulence Factors of Vibrio parahaemolyticus in the Southern Coastal Waters of Republic of Korea"

_antibiotics, 2025, doi:10.3390/antibiotics14050435_

Round 1
Reviewer 1 Report
Comments and Suggestions for Authors
This study investigates the prevalence, antibiotic resistance patterns, and virulence gene distribution of Vibrio parahaemolyticus in shellfish and seawater samples from the southern coastal waters of Korea. By integrating microbiological surveillance with environmental factors, this research offers valuable data for food safety policies and antimicrobial stewardship efforts. Below are some comments, questions, and suggestions for the improvement of the manuscript:
- Figure 1: Considering the way the y axis is written for both Figures 1A and 1B, I think the graphs could be overlapped using different colors to help the readers associate the seawater temperature and salinity.
- Discussion: Please discuss about the hemolysin toxin produced by V. parahaemolyticus and how it can be degraded. Is it degradable by heat applied in cooking? Or can it be removed by a proper cleaning?
- Discussion: Please consider and discuss the horizontal gene transfer mechanisms as a means towards the bacterial resistance pattern.
- Discussion: The results suggest a growing public health risk due to antibiotic-resistant V. parahaemolyticus. According to the authors, how should seafood safety regulations evolve based on these findings?
Thank you.
Author Response
Response to Reviewer 1
We sincerely appreciate your thoughtful and constructive comments, which greatly contributed to the improvement of our manuscript. We have carefully considered each point and revised the manuscript accordingly. Please find our point-by-point responses below.
Comment 1:
Figure 1: Considering the way the y axis is written for both Figures 1A and 1B, I think the graphs could be overlapped using different colors to help the readers associate the seawater temperature and salinity.
Response 1:
Thank you for your valuable suggestion. To improve clarity and visual comparison between seawater temperature and salinity, we have revised Figure 1 to present both variables as overlaid line graphs using distinct colors. This modification has been implemented in the revised manuscript at Line 87.
Comment 2:
Discussion: Please discuss about the hemolysin toxin produced by V. parahaemolyticus and how it can be degraded. Is it degradable by heat applied in cooking? Or can it be removed by a proper cleaning?
Response 2:
Thank you for this insightful comment. We have added a discussion on the thermal stability and inactivation of hemolysin toxins produced by V. parahaemolyticus. Specifically, we clarified that while these toxins are termed “thermostable,” they are proteinaceous and can be effectively inactivated at cooking temperatures above 70 °C, whereas rinsing or cleaning is insufficient for removal. This addition can be found at Lines 234–240.
Comment 3:
Discussion: Please consider and discuss the horizontal gene transfer mechanisms as a means towards the bacterial resistance pattern.
Response 3:
We appreciate your suggestion. In response, we have included a brief explanation regarding horizontal gene transfer (HGT) mechanisms, such as conjugation, transformation, and transduction, which may contribute to the spread of antibiotic resistance among Vibrio species in marine environments. This discussion has been added at Lines 193–198.
Comment 4:
Discussion: The results suggest a growing public health risk due to antibiotic-resistant V. parahaemolyticus. According to the authors, how should seafood safety regulations evolve based on these findings?
Response 4:
Thank you for raising this important point. We have expanded the discussion to address how the findings support the need for stronger seafood safety regulations. Specifically, we suggest incorporating antibiotic resistance surveillance into routine monitoring frameworks and implementing seasonal control measures. This response has been added at Lines 210–214.
Once again, we are grateful for your thorough review and constructive input. Your comments have been instrumental in strengthening the scientific clarity and practical relevance of our work.
Reviewer 2 Report
Comments and Suggestions for Authors
Vibrio parahaemolyticus is an important foodborne pathogen. This work investigated distribution, Antibiotic Resistance, and Virulence Factors of Vp in the Southern Coastal Waters of Korea with standard microbiological methods. Authors recovered 673 isolates were obtained from 185 shellfish and 226 seawater samples, and do a good molecular epidemiology work. The workload is impressive. Some concerns were listed here for reference.
L163: Resistance to ampicillin is caused by intrinsic resistance gene harbored by VP. Authors could refer to the reference (Chiou, J., et al. (2015). "CARB-17 family of beta-lactamases mediates intrinsic resistance to penicillins in Vibrio parahaemolyticus." Antimicrob Agents Chemother 59(6): 3593-3595). At the same time, what’s the potential mechanisms of colistin resistance? Is there any specific mechanism responsible for this high rate resistance?
L174-175: Ampicillin resistance is the intrinsic phenotype of VP, so this resistance phenotype may be not a good example to support such claim. It’s suggested to utilize the high colistin resistance case.
L245: How many colonies were isolated of each sample for following analysis? Based on the Abstract information, it seems that some VP isolates were recovered from the same sample. Authors should do a detailed introduction in the Result section. Currently, the information of 673 V. parahaemolyticus was not explained well.
L253: Did authors confirm all the 673 strains with 16s rRNA gene sequencing? This procedure actually is a little redundan.
Author Response
Response to Reviewer 2
We greatly appreciate your thoughtful review and valuable suggestions. Your comments have helped us improve the clarity and scientific rigor of our manuscript. Below, we address each of your points in detail and describe the revisions made.
Comment 1:
L163: Resistance to ampicillin is caused by intrinsic resistance gene harbored by VP. Authors could refer to the reference (Chiou, J., et al. (2015). "CARB-17 family of beta-lactamases mediates intrinsic resistance to penicillins in Vibrio parahaemolyticus." Antimicrob Agents Chemother 59(6): 3593-3595). At the same time, what’s the potential mechanisms of colistin resistance? Is there any specific mechanism responsible for this high rate resistance?
Response 1:
Thank you for pointing this out. We agree with your comment. Accordingly, we have clarified that ampicillin resistance in V. parahaemolyticus is an intrinsic trait mediated by the chromosomally encoded CARB-17 β-lactamase. Additionally, we discussed possible mechanisms of colistin resistance, such as lipid A modification via the pmr operon or mcr-like genes. This clarification has been incorporated at Lines 171–176.
Comment 2:
L174–175: Ampicillin resistance is the intrinsic phenotype of VP, so this resistance phenotype may not be a good example to support such claim. It’s suggested to utilize the high colistin resistance case.
Response 2:
We appreciate this important observation. In response, we revised the paragraph to emphasize colistin resistance as the more relevant indicator of environmental or acquired resistance. While ampicillin resistance remains discussed as an intrinsic trait, the argument for seafood safety regulation is now supported primarily by the high rate of colistin resistance. The revised content appears at Lines 180–185.
Comment 3:
L245: How many colonies were isolated of each sample for following analysis? Based on the Abstract information, it seems that some VP isolates were recovered from the same sample. Authors should do a detailed introduction in the Result section. Currently, the information of 673 V. parahaemolyticus was not explained well.
Response 3:
Thank you for your suggestion. We clarified the colony selection procedure in the methodology section. Specifically, we stated that two or three presumptive colonies were selected from each positive plate, with up to 27 colonies per sample being isolated based on the three-tube, three-dilution MPN method. This information has been added at Lines 301–304.
Comment 4:
L253: Did authors confirm all the 673 strains with 16s rRNA gene sequencing? This procedure actually is a little redundant.
Response 4:
Yes, we confirm that all 673 isolates were subjected to 16S rRNA gene sequencing. Although MALDI-TOF MS was used for preliminary species identification, 16S rRNA sequencing was conducted for all isolates to ensure accurate confirmation. Given that 16S rRNA sequencing is considered the gold standard for bacterial identification, this step was taken to minimize misidentification, particularly among closely related Vibrio species. We believe this enhances the reliability of our dataset.
Thank you again for your helpful comments. Your feedback has strengthened the manuscript both scientifically and structurally.
Reviewer 3 Report
Comments and Suggestions for Authors
This study investigated the spatiotemporal distribution, antibiotic resistance, and virulence gene profiles of Vibrio parahaemolyticus in the southern coastal waters of Korea. The substantial dataset, particularly the detailed seasonal dynamic monitoring, provides critical baseline data for regional food safety risk assessment.
The study exhibits some limitations:
- The transportation conditions (7–10°C) did not specify time limits, leaving uncertainty about potential bacterial proliferation or sample degradation during transit.
- The species’ potential intrinsic resistance to polymyxins was not experimentally ruled out. CLSI/EUCAST interpretive criteria for resistance were not explicitly cited, undermining the validity of resistance rate data.
- Sampling sites focused on shellfish farming areas (Figure 5) without including adjacent non-farming zones as controls, making it impossible to disentangle the direct contribution of aquaculture practices (e.g., antibiotic-laced feed) to elevated resistance rates.
- Although a positive association between water temperature and detection rates was noted, statistical models were not used to quantify the independent or interactive effects of temperature/salinity on bacterial abundance, resistance, or virulence genes.
Minor:
Recommendations such as “enhanced monitoring” and “antibiotic control” lack specificity.
Comments on the Quality of English LanguageFine
Author Response
Response to Reviewer 3
Thank you for your constructive feedback and thoughtful review. We have carefully considered each of your comments and revised the manuscript accordingly to improve clarity and scientific rigor. The following changes have been made in response to your suggestions:
Clarification of Sample Transportation Conditions (Lines 273–275):
We have clarified that all samples were transported under refrigeration (7–10 °C) and processed within 24 hours of collection. This addition ensures that potential bacterial proliferation or degradation during transit is minimized, thereby preserving the integrity of microbiological analyses.
Interpretive Criteria for Antimicrobial Resistance (Lines 342–348):
To address concerns regarding the interpretive criteria for resistance, we have added explicit reference to the Clinical and Laboratory Standards Institute (CLSI) M100 guidelines and the National Antimicrobial Resistance Monitoring System (NARMS) Human Isolates Report. For antimicrobials without CLSI breakpoints (e.g., streptomycin and ceftiofur), interpretations were based on NARMS 2014 criteria.
Clarification of Study Design Limitations (Lines 187–189):
We have acknowledged that the absence of non-aquaculture control sites may limit broader ecological comparisons. While our focus was on aquaculture zones due to their direct relevance to seafood safety, we have now explicitly stated this limitation in the manuscript.
Incorporation of Statistical Modeling as a Future Direction (Lines 242–244):
In response to the comment regarding statistical modeling, we have included a statement suggesting that future studies should employ quantitative models to evaluate the independent or interactive effects of environmental parameters such as temperature and salinity.
Refinement of Vague Recommendations (Lines 185–187 and 205–207):
We have revised vague terms such as “enhanced monitoring” and “antibiotic control” to specify the importance of implementing routine surveillance systems and promoting risk-based antibiotic usage regulation in aquaculture environments.
We sincerely appreciate your thorough review and helpful comments. Your suggestions have strengthened the manuscript and improved its relevance to food safety and environmental microbiology.
Reviewer 4 Report
Comments and Suggestions for Authors
The manuscript presents relevant data on Vibrio parahaemolyticus distribution, antibiotic resistance, and virulence in Korean coastal waters. The study design is appropriate and the data are well presented. However, the introduction could better emphasize what distinguishes this work from prior studies in Korea. Consider clarifying the rationale for selecting only the tdh and trh virulence genes. Please expand the discussion on the public health implications of your findings, particularly in the context of seafood safety in Korea.
Comments on the Quality of English LanguageA professional English language editing service is recommended to polish the text and ensure that your research is communicated as clearly and precisely as possible.
Author Response
Response to Reviewer 4
Thank you for your thorough review and supportive evaluation of our manuscript. We appreciate your insightful suggestions, which have helped enhance the clarity and scientific relevance of our work. In response to your comments, we have made the following revisions:
Clarification of the Study’s Distinction from Previous Research in Korea (Lines 63–77):
To better emphasize the novelty and significance of our study, we have expanded the introduction to highlight how our work differs from previous studies conducted in Korea. Specifically, we note that prior research was often limited in geographic scope or sampling duration and tended to focus solely on either antibiotic resistance or virulence. In contrast, our study offers a comprehensive year-round survey across 24 shellfish harvesting sites, integrating data on ecological distribution, resistance, and virulence gene profiles.
Justification for Selection of tdh and trh Genes (Lines 216–218):
We have added a brief explanation in the Discussion section to clarify our rationale for focusing on the tdh and trh genes. These genes were selected due to their high epidemiological relevance and their role as key molecular markers for identifying pathogenic V. parahaemolyticus.
Expansion on Public Health Implications (Lines 210–214):
The Discussion has been expanded to more explicitly address the public health significance of our findings, particularly within the context of seafood safety in Korea. We emphasize the necessity for risk-based monitoring, enhanced regulation of antimicrobial use, and targeted public health interventions during high-risk seasons.
We sincerely appreciate your thoughtful review and constructive feedback, which have contributed to improving the overall quality and impact of our manuscript.
Reviewer 5 Report
Comments and Suggestions for Authors
This study presents data on the seasonal prevalence, antimicrobial resistance, and virulence profiles of Vibrio parahaemolyticus along the Korean coastline. The strong correlation between seawater temperature and the prevalence of V. parahaemolyticus underscores the seasonal risk posed by this pathogen, particularly during warmer months. While similar studies have previously reported on the seasonal prevalence, antimicrobial resistance, and virulence profiles of V. parahaemolyticus, the primary distinction of this manuscript lies in its geographical focus rather than its biological findings.
Additional comments:
The authors reference Table 3; however, it is not included in the manuscript. Please verify and include it if necessary.
The authors should discuss why shellfish exhibited a higher overall detection rate (41.1%) compared to seawater (27.4%), yet the detection of virulence genes in shellfish was lower than in seawater.
Author Response
Response to Reviewer 5
Thank you for your valuable feedback and thoughtful evaluation of our manuscript. We appreciate your recognition of the study's strengths and the opportunity to address your comments, which have helped improve the clarity and completeness of the manuscript. Please find our detailed responses below:
Table 3 (Line 135):
We have confirmed that Table 3 was inadvertently omitted from the original submission. The missing table has now been added to the manuscript at Line 135 to ensure that the referenced data are clearly presented.
Discrepancy Between Detection Rates in Shellfish vs. Seawater (Lines 222–230):
In response to your observation regarding the higher overall detection rate of V. parahaemolyticus in shellfish (41.1%) compared to seawater (27.4%), yet a lower detection rate of virulence genes in shellfish, we have added a clarifying explanation in the Discussion. We propose that this discrepancy may be due to the relatively low overall detection rate of virulence genes in both matrices, which increases the potential impact of small sample variations. Additionally, environmental conditions such as salinity and nutrient levels may contribute to localized differences in virulence gene expression or distribution. We also acknowledge that enrichment and sampling limitations could affect these outcomes.
Thank you once again for your constructive input. Your comments have contributed meaningfully to improving the clarity and scientific rigor of our study.
Round 2
Reviewer 3 Report
Comments and Suggestions for Authors
No further comments
Reviewer 5 Report
Comments and Suggestions for Authors
Well done, the author addressed my comments very well.